# Integrating national surveys to estimate small area variations in poor health and limiting long-term illness in Great Britain

Graham Moon,[1] Grant Aitken,[2] Joanna Taylor,[3] Liz Twigg[4]

[1]Geography and Environment, University of Southampton, Southampton, UK
[2]Information Services Division, NHS National Services, Edinburgh, UK
[3]Independent Scholar, Petersfield, UK
[4]Department of Geography, University of Portsmouth, Portsmouth, UK

**Correspondence to**
Dr Graham Moon;
g.moon@soton.ac.uk

## ABSTRACT

**Objectives** This study aims to address, for the first time, the challenges of constructing small area estimates of health status using linked national surveys. The study also seeks to assess the concordance of these small area estimates with data from national censuses.

**Setting** Population level health status in England, Scotland and Wales.

**Participants** A linked integrated dataset of 23 374 survey respondents (16+ years) from the 2011 waves of the Health Survey for England (n=8603), the Scottish Health Survey (n=7537) and the Welsh Health Survey (n=7234).

**Primary and secondary outcome measures** Population prevalence of poorer self-rated health and limiting long-term illness. A multilevel small area estimation modelling approach was used to estimate prevalence of these outcomes for middle super output areas in England and Wales and intermediate zones in Scotland. The estimates were then compared with matched measures from the contemporaneous 2011 UK Census.

**Results** There was a strong positive association between the small area estimates and matched census measures for all three countries for both poorer self-rated health (r=0.828, 95% CI 0.821 to 0.834) and limiting long-term illness (r=0.831, 95% CI 0.824 to 0.837), although systematic differences were evident, and small area estimation tended to indicate higher prevalences than census data.

**Conclusions** Despite strong concordance, variations in the small area prevalences of poorer self-rated health and limiting long-term illness evident in census data cannot be replicated perfectly using small area estimation with linked national surveys. This reflects a lack of harmonisation between surveys over question wording and design. The nature of small area estimates as 'expected values' also needs to be better understood.

## Strengths and limitations of this study

► Reports on small area estimates of health status derived from the novel linkage of three large well-designed routine national health surveys.
► Develops and enhances a well-reputed approach to the small area estimation of health indicators.
► Highlights the enduring challenges entailed in harmonising measures of health status across different surveys.
► Enhances understanding of the reasons that underlie discrepancies between small area estimates and equivalent observed data.

robust and reliable standards that are shared nationally and, ideally, internationally. Moreover, they need to capture the health status of whole populations, to ensure comprehensive rather than selective coverage. These ideals are seldom met. National surveys are not usually designed to provide valid data down to suitably small areas, although an exception is offered in the few cases where national censuses provide small area health data. Local surveys, if they exist at all, can vary substantially in design, making comparisons difficult. Using administrative data to make local estimates restricts attention to users of services and thus lacks population representativeness.

To address the unmet need for small area data, researchers across the world have increasingly turned to small area estimation (SAE) methods.[2] SAE methods use statistical or mathematical methods to manipulate national survey data to produce local estimates for a target measure. A common methodological root is the association in a national survey between a target variable of interest and covariates thought to predict that target variable. Estimation uses local data on the covariates. The SAE process generally takes place within a single national setting

## INTRODUCTION

Small area data on variations in health status have long been seen as central to geographical comparisons of health needs.[1] Effectiveness is enhanced when data are available at very local scales facilitating sensitive, community-focused action. Such data additionally need to be provided to consistent,

with multiple surveys being used to enhance the pool of available covariates. There has been limited attention to linking surveys from different geographical contexts to enhance the spatial coverage of estimates through a larger pool of cases. Addressing this omission recognises that health issues seldom respect geographical borders. It also enables the identification of 'place effects' on outcome measures and reveals the extent to which definitions of key health variables differ with geographical setting.

Health applications of SAE have focused mainly on outcomes and behaviours.[3–7] Rather, less attention has been given to SAEs of general measures of health status such as limiting long-term illness (LLTI) and self-reported general health (SRGH). Such measures are good at picking up ageing populations and chronic illness as well as pockets of severe health deprivation.[8–13] For these reasons, they are frequently included in national censuses. Although this might be held to obviate the need for SAEs of health status, it presents an opportunity to address a frequent criticism of SAEs: the absence of validation. Small area census data on health status provide what is arguably a 'gold standard' against which SAEs can be compared.[14]

Motivated by the lack of previous research on health-related SAE using surveys from more than one geographical setting and the opportunity to ground-truth SAEs of health status using small area census data, this paper addresses two objectives. First, we develop parsimonious SAE models of LLTI and SRGH using surveys from multiple geographical settings. Second, the estimates from these models are compared with population census data. As both SAEs and census data are population-level measures, we hypothesise a close concordance.

## METHODS

We created SAEs by modelling routine data and applying the coefficients from our models to census data on the covariate measures. Our SAEs were then compared with census data on LLTI and SRGH. Great Britain was the setting for the study as we had ready access to the separate surveys for the constituent countries of Wales, Scotland and England as well as to census data for each country. LLTI and SRGH are established measures of health status in each country. The UK Census began collecting information on SRGH in 2001 and LLTI in 1991.

### Small area estimation

SAE models were derived primarily from a linked data file comprising the individual responses to the 2011 versions of the Health Survey for England (HSfE), the Scottish Health Survey (SHS) and the Welsh Health Survey (WHS). These sources were accessed through the UK Data Service. Surveys from 2011 were selected to facilitate direct comparison with the 2011 decennial population census. Full details of the conduct of the surveys, the measurement of variables and response rates are given in the 2011 HSfE, SHS and WHS reports.[15–17] Collectively, the sources offer comprehensive well-found authoritative insights into health status in each country. After data cleaning to address missing data, the working data file comprised 23 374 individuals (aged 16+ years; England n=8603, Scotland n=7537, Wales n=7234).

Data from the three health surveys were supplemented with linked data on disability and multiple deprivation at the area level, in recognition of the close association between these factors and our target variable.[8 10 18] We obtained special permission to use geocoded data to link these additional variables for middle (layer) super output areas (MSOAs). MSOAs (known as intermediate zones in Scotland, but referred to as MSOAs throughout this paper) total 6791 in England, 1235 in Scotland and 410 in Wales. They are small areas ranging in population from 5000 to 15 000 in England and Wales and 2500 to 6000 in Scotland. We added the combined rate of disability living allowance and attendance allowance per 1000 adults per MSOA, and the combined rate of incapacity benefit plus severe disablement allowance plus employment and support allowance per 1000 adults per MSOA. Both benefit measures were derived from data held on the NOMIS website.[19] For deprivation, we used a UK-wide Index of Multiple Deprivation (IMD) score, calibrated from the English, Scottish and Welsh versions of the index using a modified version of the method outlined by Payne and Abel.[20] To track survey effects, we also added a flag denoting whether a respondent was from England, Scotland or Wales.

The outcome measures for our two SAE models were poorer SRGH and possession of an LLTI. Both measures required harmonisation of the relevant questions across the three surveys and in the decennial population census (table 1). For SRGH, respondents were asked to self-report their general health. In the census, HSfE and SHS 'very good' or 'good' general health was coded as good SRGH; 'fair', 'bad' or 'very bad' general health was coded as poorer SRGH. In the WHS 'excellent', 'very good' or 'good' general health was coded as good SRGH, and 'fair' or 'poor' general health was coded as poorer SRGH. LLTI was also dichotomised. In the census and the WHS, respondents were asked first to define if they had any long-term illness, health problem or disability which limited their daily activities or the work they could do. If they responded 'yes, limited a lot' or 'yes, limited a little', they were coded as having a LLTI, and if they responded 'no', they were not. In the HSfE, people were asked if they had a long-standing illness, disability or infirmity troubling them over a period of time. If they responded yes, they were then asked if this limited their activities in any way, with those individuals answering yes to this second question being coded as having an LLTI. In the SHS, people were if they had a long-standing physical or mental condition or disability that troubled them for at least 12 months. If they responded yes, they were asked if this condition limited their activities in any way, with those individuals answering yes to this question coded as having an LLTI.

**Table 1** Wording of SRGH and LLTI questions for Census, Health Survey for England, the Scottish Health Survey and the Welsh Health Survey for people aged 16+

**SRGH: question and responses**

| | | | | |
|---|---|---|---|---|
| 2011 Census | How is your health in general? | Very good or good | Fair | Bad or very bad |
| 2011 Health Survey for England | How is your health in general? Would you say it was… | Very good or good | Fair | Bad or very bad |
| 2011 Scottish Health Survey | How is your health in general? Would you say it was… | Very good or good | Fair | Bad or very bad |
| 2011 Welsh Health Survey | In general, would you say your health is? | Excellent, very good or good | Fair | Poor |

**LLTI: question and responses**

| | | | | |
|---|---|---|---|---|
| 2011 Census | Are your day-to-day activities limited because of a health problem or disability which has lasted, or is expected to last, at least 12 months? Include problems related to old age. | No | Yes, limited a little | Yes, limited a lot |
| 2011 Health Survey for England | Do you have any long-standing illness, disability or infirmity? By long-standing, I mean anything that has troubled you over a period of time or that is likely to affect you over a period of time? | No | Yes | |
| | If yes, does this illness or disability/do any of these illnesses or disabilities limit your activities in any way? | No | Yes | |
| 2011 Scottish Health Survey | Do you have a long-standing physical or mental condition or disability that has troubled you for at least 12 months, or that is likely to affect you for at least 12 months? | No | Yes | |
| | If yes, does (name of condition) limit your activities in any way? | No | Yes | |
| 2011 Welsh Health Survey | Are your day-to-day activities limited because of a health problem or disability which has lasted, or is expected to last, at least 12 months? Include problems related to old age. | No | Yes, limited a little | Yes, limited a lot |

LLTI, limiting long-term illness; SRGH, self-reported general health.

Covariate data to facilitate the generation of small area estimates were derived from the linked data file. A parsimonious selection of covariates was made reflecting the known associations between LLTI and SRGH and age, sex, disability and deprivation.[21 22] Individual-level age and sex were self-reported measures. Age was grouped into 13 categories: those aged 16–19, 5-year age groups until the age of 74 and all individuals aged 75+. These individual-level covariates were supplemented with the area-level measures of disability and deprivation introduced above.

To create SAEs, we used the established multilevel small area estimation process recognising the hierarchical structure of the source surveys.[3 23 24] This approach involves developing a multilevel model using survey data with covariate terms that are also available for all target small areas. The process began with initial data management into individual and area-level covariates using IBM SPSS Statistics V.22. Area-level variables were centred on their grand mean. We then developed two-level logistic models of individuals nested within MSOAs using MLwiN (V.2.35).[25 26] Separate models were produced to estimate SRGH and LLTI. Models were tested for interactions between the age and sex terms, and variables were retained in the final models if they were found to be statistically significant using the $X^2$ test (p≤0.05). We considered modelling with the individual non-response weights available in each of the national surveys but, after exploration, elected to proceed with unweighted data. Views vary on whether to use weighted data in small area estimation with the Bayesian nature of our modelling process offering support for our decision.[27]

Our SAE models were initially estimated using iterative generalised least squares (IGLS) with first order maximum quasi-likelihood estimation and took the general form:

$$Y_{ij} = \beta_{0j} + \boldsymbol{\beta_{1ij}} + \boldsymbol{\beta_{2j}} + \mu_{0j}$$

where Y represents the outcome, whether an individual has an LLTI or is in poor health, $\beta_{0j}$ is the intercept in the model, $\boldsymbol{\beta_{1ij}}$ represents covariates measured at the individual level (age and sex), $\boldsymbol{\beta_{2j}}$ refers to covariates measured at the MSOA level (benefit receipts, IMD and the flag denoting England, Scotland or Wales), and $\mu_{0j}$ indicates the MSOA level variance; individual-level variance is constrained to one in the binomial model. Once the IGLS models achieved convergence, their coefficients were used as informative priors in Bayesian Monte Carlo Markov Chain (MCMC) models to allow for more robust estimates and SEs. Both MCMC models were run through 500 000 iterations, with an initial burn-in period of 50 000 iterations. The SAE process concluded with the generation of SAEs of SRGH and LLTI at MSOA level produced by converting the final MCMC model logit coefficients to probabilities and applying them to a data file for all MSOAs in Great Britain comprising cross-tabulations of age and sex from the 2011 Decennial Population Census together with the area-level indicators.

## Census comparison

In the UK, the 2011 Census is the most recent source of information on local inequalities in SRGH and LLTI. We compared our MSOA SAEs to census data on SRGH and LLTI for MSOAs. For England and Wales, this was sourced from the ONS official labour market statistics website. For Scotland, it was downloaded from the Scottish Census data warehouse.

SAEs and census estimates were compared using regression and correlation analysis following the Scarborough methodology used previously for validating SAEs for coronary heart disease.[28] SAEs were plotted against the census measurement of the same target variable at the MSOA level. Convergent validity was achieved if the line of best fit had a gradient with CIs including one and an intercept with CIs including zero. For each model, we considered four regression lines: one for Great Britain as a whole and one each for the constituent countries of England, Scotland and Wales. We also sought high correlations.

## Results

A descriptive summary of the full linked data file used to construct the SAEs is set out in table 2. The resulting small area estimation models are shown in table 3. We show logits and SEs as these provide the input to the SAE process. Having poor SRGH was largely a function of increasing age and higher MSOA IMD scores (increased deprivation). A similar picture was evident for LLTI with being female playing an additional role. MSOA-level benefit measures had little effect on either SRGH or LLTI. For SRGH, the models showed no difference between Scotland and the reference country of England but a markedly lower likelihood of poorer SRGH in Wales. In contrast, being located in either Wales or Scotland was associated with a higher likelihood of reporting an LLTI compared with England. Both models were relatively successful in capturing variation in their outcome measures, with the LLTI model being marginally more effective.

Summary statistics on the SAEs derived from these models are presented for England, Scotland and Wales in table 4. The mean MSOA prevalence of poorer SRGH is highest in Scotland and lowest in Wales. For LLTI, Wales has the highest mean MSOA prevalence, and England is lowest. In both Scotland and Wales, the mean MSOA prevalence for LLTI is notably higher than that for poorer SRGH. Table 4 also indicates the mean MSOA census prevalences for poorer SRGH and LLTI for each country, providing initial insights into the match between SAEs and 'gold standard' census data. Differences are evident. For poorer SRGH, Scotland has the lowest mean prevalence on the Census measure but the highest on SAE measure; for Wales, the situation is reversed. Both England and Scotland have a lower mean prevalence on the census measure compared with the SAE, while for Wales, the SAE prevalence is higher. In the case of LLTI, the relative position of the three countries is the same for

**Table 2** Descriptive statistics: full linked dataset

| Variable | Category | Full sample | | General health | | LLTI | |
|---|---|---|---|---|---|---|---|
| | | Number | Column % | Number | Column % | Number | Column % |
| All | Aged 16+ | 23 374 | 100 | 23 355 | 100 | 23 281 | 100 |
| Country | England | 8603 | 36.8 | 8603 | 36.8 | 8603 | 37.0 |
| | Scotland | 7537 | 32.3 | 7537 | 32.3 | 7537 | 32.4 |
| | Wales | 7234 | 30.9 | 7215 | 30.9 | 7141 | 30.5 |
| Sex | Male | 10 285 | 44.0 | 10 276 | 44.0 | 10 240 | 44.0 |
| | Female | 13 089 | 56.0 | 13 079 | 56.0 | 13 041 | 56.0 |
| Age | 16–19 | 772 | 3.3 | 770 | 3.3 | 780 | 3.4 |
| | 20–24 | 1146 | 4.9 | 1146 | 4.9 | 1153 | 5.0 |
| | 25–29 | 1403 | 6.0 | 1 403 | 6.0 | 1404 | 6.0 |
| | 30–34 | 1650 | 7.1 | 1650 | 7.1 | 1645 | 7.1 |
| | 35–39 | 1790 | 7.7 | 1788 | 7.7 | 1783 | 7.7 |
| | 40–44 | 2123 | 9.1 | 2122 | 9.1 | 2112 | 9.1 |
| | 45–49 | 2140 | 9.2 | 2138 | 9.2 | 2136 | 9.2 |
| | 50–54 | 2032 | 8.7 | 2030 | 8.7 | 2023 | 8.7 |
| | 55–59 | 1941 | 8.3 | 1940 | 8.3 | 1935 | 8.3 |
| | 60–64 | 2272 | 9.7 | 2271 | 9.7 | 2268 | 9.7 |
| | 65–69 | 1917 | 8.2 | 1914 | 8.2 | 1907 | 8.2 |
| | 70–74 | 1509 | 6.5 | 1508 | 6.5 | 1494 | 6.4 |
| | 75+ | 2679 | 11.4 | 2675 | 11.4 | 2641 | 11.3 |
| General health | Fair/poor/very poor | – | – | 5956 | 25.5 | – | – |
| | Very good/good | – | – | 17 399 | 74.5 | – | – |
| LLTI | Present | – | – | – | – | 7309 | 31.4 |
| | Not present | – | – | – | – | 15 972 | 68.6 |
| | | Mean (SD) | | Mean (SD) | | Mean (SD) | |
| Disability benefits | | 109.39 (±45.19) | | 109.38 (±45.20) | | 109.18 (±45.13) | |
| Work benefits | | 86.66 (±57.49) | | 86.68 (±57.51) | | 86.73 (±57.60) | |
| IMD | | 19.11 (±10.67) | | 19.10 (±10.67) | | 19.07 (±10.67) | |

IMD, Index of Multiple Deprivation; LLTI, limiting long-term illness; SRGH, self-reported general health.

the two measures, but the census mean prevalences are substantially lower.

The concordance between SAEs and census data is explored further in figure 1. The diagonal reference line captures the scenario where the SAE and census estimates would match. The SAEs for poorer SRGH in England and Scotland were higher for most small areas compared with the census, while the great majority of Welsh estimates were lower. For LLTI, most SAE prevalences exceeded the equivalent census prevalences; those MSOAs with SAEs less than the corresponding census values tended to be in England.

Table 5 continues the exploration of the match between SAEs and census using the 'Scarborough criteria'. There are strong correlations between the two measures across all settings suggesting broad agreement. However, neither for individual countries nor collectively do any of the SAEs exhibit anything approaching the requirements for a strong concordance with census measures. All have

intercepts significantly above zero and gradients that depart markedly from one.

## DISCUSSION

Even with jurisdictions as closely associated as England, Scotland and Wales and ongoing UK-wide attempts at harmonising questions in health and other routine national surveys, there remain differences in wording and format that pose difficulties for SAE when it comes to linking survey input data across geographical settings. Overcoming these difficulties requires compromise and accommodation but is possible. In response to our first objective, we have shown how a cross-setting dataset can be developed as the basis for a parsimonious SAE model reflecting the key social determinants of poorer SRGH and LLTI.

Our second objective entailed comparing our SAEs with matched and contemporaneous MSOA census data.

**Table 3** SRGH and LLTI model parameters: logit and SE values

|  | Poorer SRGH model | | LLTI model | |
|  | Logit | SE | Logit | SE |
|---|---|---|---|---|
| Constant | −2.470 | 0.137 | −2.422 | 0.122 |
| Sex: female | 0.059 | 0.032 | 0.151 | 0.031 |
| Age: 20–24 | 0.188 | 0.165 | −0.193 | 0.156 |
| Age: 25.29 | 0.339 | 0.157 | 0.178 | 0.143 |
| Age: 30–34 | 0.208 | 0.156 | 0.173 | 0.140 |
| Age: 35–39 | 0.680 | 0.148 | 0.455 | 0.135 |
| Age: 40–44 | 0.863 | 0.144 | 0.654 | 0.131 |
| Age: 45–49 | 1.168 | 0.142 | 0.984 | 0.129 |
| Age: 50–54 | 1.465 | 0.142 | 1.281 | 0.128 |
| Age: 55–59 | 1.650 | 0.142 | 1.436 | 0.128 |
| Age: 60–64 | 1.758 | 0.140 | 1.671 | 0.127 |
| Age: 65–69 | 1.886 | 0.141 | 1.816 | 0.128 |
| Age: 70–74 | 2.086 | 0.143 | 2.023 | 0.130 |
| Age: 75+ | 2.541 | 0.138 | 2.617 | 0.126 |
| IMD | 0.023 | 0.003 | 0.011 | 0.003 |
| Disability benefits | 0.002 | 0.001 | 0.002 | 0.001 |
| Work benefits | −0.001 | 0.001 | - | - |
| Scotland | 0.150 | 0.110 | 0.319 | 0.046 |
| Wales | −0.456 | 0.056 | 0.469 | 0.049 |
| Deviance information criterion reduction* | 2179 | | 3031 | |
| Explained variance† | 15.6% | | 19.5% | |

Greyed out data are not statistically significant (p>0.05; Wald test). The models' constants are a man aged 16–19 years, living in an MSOA of average IMD, in disability receipt and work benefit receipt, and resident in England.
*Deviance information criterion reductions from null models. Larger reductions indicate better models.[29]
†Computed following Snijders and Bosker latent variable approach.[30]
IMD, Index of Multiple Deprivation; LLTI, limiting long-term illness; SRGH, self-reported general health.

Here, our results were at best equivocal. While strong correlations were evident, suggesting a tight association between the two measures running in an expected direction, closer inspection revealed that SAEs tended broadly to exceed census estimates, and MSOAs in different national contexts returned starkly varying results across the two measures. It would, of course, be possible, in the case of this specific example, to go further and weight the SAEs with the census data to bring the SAEs closer to the presumed (census) gold standard. More generally however, SAEs are created to fill the gaps occasioned by the absence of gold standard data at a local geographical scale, meaning post hoc adjustment at a local level is seldom possible. It can though be done when gold standard data are available at a higher spatial level, ensuring that local SAEs sum to known higher level figures.

Despite this lack of concordance, our findings update previous knowledge on SRGH and LLTI in Great Britain. Levels of LLTI were found to be higher in Wales than in Scotland or England using the 1991 Census data.[31] This

**Table 4** Middle super output areas level SAEs and census estimates compared

|  | Poorer SRGH | | | | | LLTI | | | | |
|  | Census mean | SAE | | | | Census mean | SAE | | | |
|  |  | Mean | SD | Min | Max |  | Mean | SD | Min | Max |
|---|---|---|---|---|---|---|---|---|---|---|
| England | 22.04 | 25.92 | 5.53 | 10.64 | 55.76 | 21.48 | 25.23 | 4.09 | 9.4 | 43.96 |
| Scotland | 21.33 | 29.17 | 6.59 | 11.89 | 54.62 | 23.10 | 31.47 | 4.86 | 14.31 | 48.33 |
| Wales | 26.23 | 21.65 | 5.23 | 6.68 | 41.36 | 27.76 | 37.29 | 4.97 | 13.97 | 51.54 |

All figures are expressed as percentages except SD.
LLTI, limiting long-term illness; SAE, small area estimation; SRGH, self-reported general health.

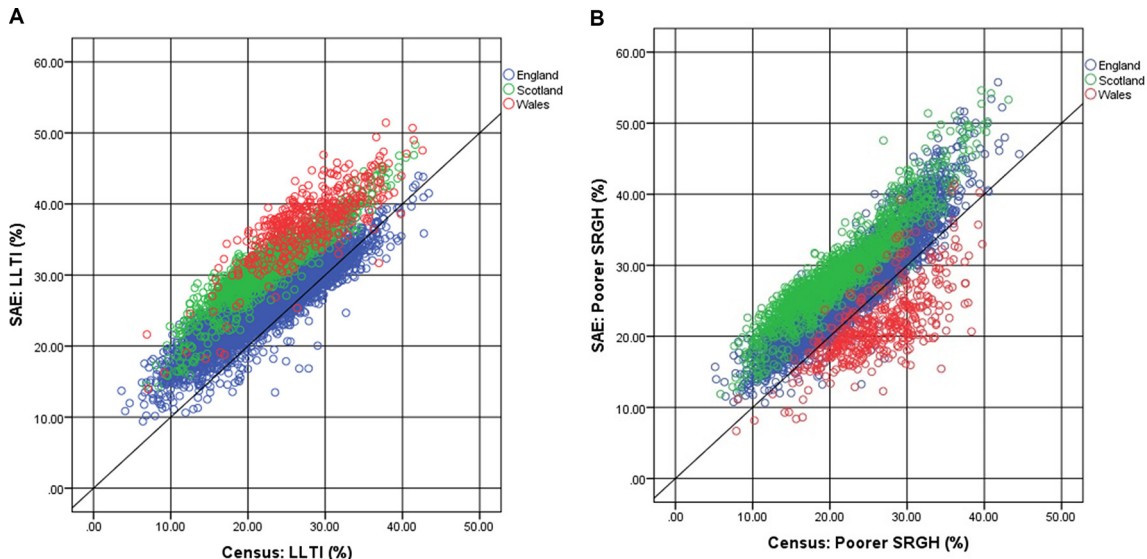

**Figure 1** Small area estimations versus census estimates: limiting long-term illness (A) and poorer self-reported general health (B).

situation still persists and is evident in our SAEs, which run counter to suggestions that models underpredict LLTI in Wales but possibly confirm a continued tendency to overprediction in Scotland.[32] The 2001 Census data was used to highlight higher rates of poor SRGH in Wales compared with the rest of Great Britain.[33] Another study with the 2001 Census data confirmed that levels of poorer SRGH were the worst in Wales, followed by Scotland, then England.[34] Our analysis sustains this finding with 2011 Census data, but our SAEs suggest a contrary picture in which Wales has better levels of poorer SRGH than either Scotland or England.

Reasons for the differences between our SAEs and contemporaneous census data for matched measures are of relevance to future research on SAE and merit examination. Differences in the wording of the SRGH and LLTI questions were evident between the source surveys and between the surveys and the census. There were also variations in the categorisation of outcome possibilities and in the positioning of the questions within the surveys. In the English and Scottish surveys the SRGH question appeared at the start of the individual interview immediately after age, with the LLTI question appearing immediately afterwards. In the WHS, additional health questions were asked before the SRGH questions. Asking about SHRG after, rather than before, other health questions, results in more positive health assessments and replacing 'very good' with 'excellent' as a response category has a similarly more positive effect.[35] The wording of the WHS question about LLTI and specifically the inclusion of the reference to 'problems of old age' may have led to greater levels of problem identification.[9 11] Collectively, these

**Table 5** Convergent validity of SAEs against census estimates at MSOA level

|  | Intercept | Lower CI | Upper CI | Contain zero? | Gradient | Lower CI | Upper CI | Contain one? | Correlation |
|---|---|---|---|---|---|---|---|---|---|
| **Poorer SRGH** | | | | | | | | | |
| GB | 6.87 | 6.58 | 7.16 | X | 0.87 | 0.86 | 0.89 | X | 0.828** |
| England | 4.98 | 4.75 | 5.21 | X | 0.95 | 0.94 | 0.96 | X | 0.916** |
| Scotland | 8.64 | 8.18 | 9.10 | X | 0.96 | 0.94 | 0.98 | X | 0.933** |
| Wales | 3.76 | 2.04 | 5.49 | X | 0.68 | 0.62 | 0.75 | X | 0.719* |
| **LLTI** | | | | | | | | | |
| GB | 9.24 | 8.99 | 9.50 | X | 0.79 | 0.78 | 0.81 | X | 0.832** |
| England | 10.56 | 10.36 | 10.74 | X | 0.68 | 0.67 | 0.69 | X | 0.895** |
| Scotland | 13.31 | 12.87 | 13.74 | X | 0.79 | 0.77 | 0.81 | X | 0.922** |
| Wales | 17.98 | 16.48 | 19.47 | X | 0.69 | 0.64 | 0.75 | X | 0.789* |

*Significant p<0.05.
**Significant p<0.01.
GB, Great Britain; LLTI, limiting long-term illness; MSOA, middle super output areas; SAE, small area estimation; SRGH, self-reported general health.

points echo calls from other jurisdictions for continued work on the harmonisation of questions between surveys and censuses across national borders.[36]

Another reason for discrepancies between our SAEs and census data is that census and survey information are collected in different ways. The householder is responsible for ensuring that the UK Census is completed, whereas the surveys are completed by an individual. Furthermore, the UK Census is a self-completion form, but our surveys were interviewer administered. Both these factors can lead to census prevalence data generally being lower than that from surveys.[37] One manifestation of both this and the previous point is that the input survey data for our SAE models (table 2) collectively evidenced prevalences of 25.5% for poorer SRGH and 31.4% for LLTI, both greater than all bar one of the corresponding national prevalences evident for census data in table 3. Inevitably, the SAEs accord more with these input data.

Although soundly based on theory, our models were undeniably simple. Research using a multinomial LLTI outcome measures has found improved concordance with the 2011 Census estimates at MSOA level in England.[38] Additional socioeconomic, health and clinical covariates, known to be associated with increased SRGH or LLTI prevalence, were not included within the modelling framework due to data constraints and the requirements of the SAE process. These omissions may have been significant and uncaptured by the areal IMD and benefit measures. For example, there may be systematic differences between social groups in their understanding of SRGH, and the omission of ethnicity is undoubtedly significant.[39]

As a final point, we reflect on the nature of SAEs and census data. SAEs are perhaps most usefully seen as expected values for the target variable, given the covariates included in the modelling process.[40 41] Census data are also estimates, subject to differential response and variations in understanding.[42 43] The census is thus not a 'gold standard', although it comes close. We should not anticipate that SAEs should have to match exactly to the estimates provided by their equivalent census measures. Rather, SAEs provide an expectation against which census data can be compared. In our analysis, linking surveys across three countries thus points to an expectation that levels of poorer SRGH and LLTI should be rather higher than suggested by census data, given variations in population distributions by age and sex and the varying areal prevalences of benefit take-up and deprivation.

**Contributors** GM and LT conceived and led the study and participated in data analysis and interpretation. JT led on data acquisition and participated in data analysis. GA participated in data analysis and interpretation. All authors contributed to drafting, revision and final approval of the manuscript. All authors are responsible for the manuscript.

**Funding** This work was supported by the UK Economic and Social Research Council grant number ES/K003046/1.

**Competing interests** None declared.

**Patient consent** This paper analyses secondary data from the UK Data Service.

**Ethics approval** University of Southampton, Faculty of Social, Human and Mathematical Sciences Research Ethics Committee.

**Provenance and peer review** Not commissioned; externally peer reviewed.

**Data sharing statement** Data files are available by emailing Graham Moon (g. moon@soton.ac.uk)

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
