## [Reviewer comments · BMJ Open]

ARTICLE DETAILS

TITLE (PROVISIONAL)	Integrating national surveys to estimate small area variations in poor health and limiting long term illness in Great Britain
AUTHORS	Moon, Graham; Aitken, Grant; Taylor, Joanna; Twigg, Liz

VERSION 1 - REVIEW

REVIEWER	Hongjian Yu Hongjian Yu, Ph.D. Statistical Unit Director UCLA Center for Health Policy Research 10960 Wilshire Blvd., Suite 1550 Los Angeles, CA 90024, USA
REVIEW RETURNED	10-Apr-2017

GENERAL COMMENTS	I recommend publishing this paper with revision to address the following issues: 1. Given the simplicity of the models, how do authors deal with weights from multiple surveys? This issue is also related to design and model consistency of the estimated model coefficients. (Pfeffermann 1998, Grilli and Pratesi 2004).2. It is a pleasant surprise to see estimates from SAE correlated with census so well! If authors think that census is the gold standard, use it to calibrate SAE estimates may bring these two sets in line.3. Some model description (equation) would be helpful for better understanding of the model. Minor comments: • It is a little difficult to understand the geography for readers who are not familiar with UK system.• The meaning of the sentence on page 15, line 17-25, is not clear to me.
---

REVIEWER	NoelleAngelique Molinari Centers for Disease Control and Prevention United States
REVIEW RETURNED	21-Apr-2017

GENERAL COMMENTS	This is a very good and interesting manuscript. The work is clearly written and well described with just a few exceptions. The Methods should be more clearly described and provide more detail to ensure transparency. On page 7, lines 17 through 46, the multilevel estimation methods are not sufficiently described. In particular, there is no description of the "multilevel" results and, while it appears Bayesian methods were used (see credible intervals" on line 46), these are not described at
--

	all. It is not sufficient to provide a citation. Lines 36 to 39, mention retaining variables based on Chi-squared tests but not the levels of significance used as criteria. One Page 10, Table 3 presents results of these estimations but does not provide any measure of fit or any diagnostics. It would be worthwhile to compare several specifications for performance. "Logit" is assumed to refer to the parameter estimates of regression coefficients. Including significance of these results would be convenient. Is it possible to include age as a continuous variable in these regressions? Why not try multinomial logistic regressions and compare performance of these to the binomial logit? This may prove especially helpful given the Census estimates are not truly well aligned with SAEs.
--	---

VERSION 1 – AUTHOR RESPONSE

Reviewer 1

1. Weights. This is an important point on which there is significant debate, including within small area estimation. We explored weighting when developing the models and chose to proceed to with unweighted data as our modelling strategy is Bayesian and should produce robust standard errors. We have added text to clarify this issue (p7)
2. Using census data to calibrate SAE estimates to bring these two sets in line. This would indeed be possible but only in this case because census data are available at a local level. This is not normally the case with SAEs but we have added text on this important possibility (p14)
3. Some model description (equation) would be helpful for better understanding of the model. We have added a generalised equation summarising our models (p7)
4. It is a little difficult to understand the geography for readers who are not familiar with UK system. We have added detail on MSOA sizes and made it clear that England, Scotland and Wales are the constituent countries of Great Britain.
5. The meaning of the sentence on page 15, line 17-25, We have given the paper a thorough reading to ensure clarity and made several minor amendments..

Reviewer: 2

1. Small area estimation method. We have added a sentence describing the multilevel small area estimation method and amended the following paragraphs to make clear that they describe the SAE process. (p7-8)
 2. Multilevel results. We have inserted a pointer on p8 to Table 3 where the multilevel model results are reported. The first paragraph of the Results section presents a description of the multilevel results.
 3. Bayesian methods. We have edited p8 to add further detail on our use of Bayesian MCMC methods. The reference to credible intervals has been removed as it is not relevant.
 4. Chi Square tests. We have added the level of significance criterion
 5. Table 3. We have added indications of the significance of the parameter estimates and two measures of fit, commenting on the latter in the text
- Referee 2 also suggested alternative approaches to modelling. We did not have age as a continuous variable so were unable to use it our modelling. With regard to possible multinomial modelling, our focus in this paper was on poorer self-reported health and having a limiting long-term illness; for this reason we used binomial models. The different question wordings set out in Table 1 also pointed us in this direction and the limiting long-term illness measure was only available as a dichotomy.

VERSION 2 – REVIEW

REVIEWER	Hongjian Yu UCLA, USA
REVIEW RETURNED	31-May-2017

GENERAL COMMENTS	Even though it seems sufficient for this publication, I wished the authors could address more in the effect of sample design and/or probability of selection from multiple surveys on the resulting SAE.
--